# FastDINOv2: Frequency Based Curriculum Learning Improves Robustness and Training Speed

**Jiaqi Zhang**[1]    **Juntuo Wang**[1]    **Zhixin Sun**[2]    **John Zou**[1]    **Randall Balestriero**[1]

[1]Brown University    [2]Cornell University
{jiaqi_zhang6, juntuo_wang, john_zou, randall_balestriero}@brown.edu
zs454@cornell.edu

## Abstract

Large-scale vision foundation models such as DINOv2 boast impressive performances by leveraging massive architectures and training datasets. But numerous scenarios require practitioners to reproduce those pre-training solutions, such as on private data, new modalities, or simply for scientific questioning–which is currently extremely demanding computation-wise. We thus propose a novel pre-training strategy for DINOv2 that simultaneously accelerates convergence–and strengthens robustness to common corruptions as a by-product. Our approach involves a frequency filtering curriculum–low-frequency being seen first–and the Gaussian noise patching augmentation. Applied to a ViT-B/16 backbone trained on ImageNet-1K, while pre-training time and FLOPs are reduced by $1.6\times$ and $2.25\times$, our method still achieves matching robustness in corruption benchmarks (ImageNet-C) and maintains competitive linear probing performance compared with baseline. This dual benefit of efficiency and robustness makes large-scale self-supervised foundation modeling more attainable, while opening the door to novel exploration around data curriculum and augmentation as means to improve self-supervised learning models robustness. The code is available at https://github.com/KevinZ0217/fast_dinov2

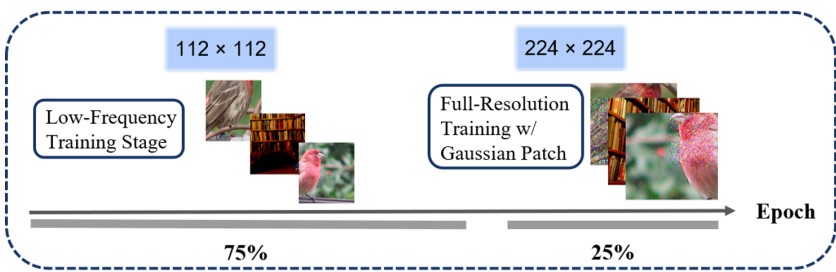

Figure 1: The FastDINOv2 training pipeline comprises two stages. In the first stage, the initial 75% of training epochs utilize only low-frequency features extracted via downsampling. In the second stage, the remaining 25% of epochs employ full-resolution images with Gaussian noise patching.

Table 1: Comparison of training costs, evaluation on ImageNet-C, and linear probing on the ImageNet-1K validation set using a frozen ViT-B DINOv2 backbone trained on ImageNet-1K.

| Training Method | Training Time (days)($\downarrow$) | ImageNet-1K($\uparrow$) | ImageNet-C ($\downarrow$) | GFLOPs ($\downarrow$) |
|---|---|---|---|---|
| DINOv2 | 16.64 (NVIDIA L40S) | 77.8% | 56.5% | 493.76 |
| **FastDINOv2** | **10.32 (NVIDIA L40S)** | **76.2**% | **56.7**% | **219.92** |

39th Conference on Neural Information Processing Systems (NeurIPS 2025).

# 1    Introduction

Models such as DINOv2 [22] and CLIP [25], built on Vision Transformer (ViT) backbones [11], achieve remarkable performance, generalization, and even upstream robustness  [39] [24]. These advances stem largely from self-supervised learning (SSL)—a paradigm in which models learn useful representations from unlabeled data by solving proxy tasks (e.g., contrastive matching, predicting masked patches) rather than relying on costly manual annotation [12]. SSL's appeal lies in its ability to leverage the massive, ever-growing pools of raw images available online; by discovering structure in data itself, SSL yields features that transfer effectively to downstream tasks such as classification, objection detection, and segmentation, often matching or exceeding their fully supervised counterparts [9] [13].

However, reproducing these recent breakthroughs typically demands huge compute: large ViT variants (e.g., ViT-G) trained for hundreds of epochs on billions of images [4], multi-GPU clusters, and sophisticated optimization recipes. This resource barrier can put state-of-the-art SSL out of reach for many academic labs and startups, limiting both reproducibility and further innovation. Moreover, while pre-trained SSL models frequently demonstrate excellent performance on clean benchmarks, robustness to real-world distribution shifts—common corruptions, sensor noise, weather effects—remains crucial for safety-critical applications such as medical imaging [2] [30]or autonomous driving [4]. Although some studies have shown that SSL can improve robustness compared to supervised pre-training [18] [3] [21], most SSL methods do not explicitly optimize for robustness, and existing robust pre-training often compounds the compute demands [26] [35].

In particular, recent large-scale SSL models exhibit an emergent robustness only when trained at extreme scales: for instance, DINOv2 and related methods require model sizes upward of 86 millions of parameters for ViT-B and datasets of similar magnitude, such as LVD-142M and LAION-5B [28], to yield strong resistance to corruptions. This scale-driven robustness is appealing in principle, but is prohibitively expensive in practice for researchers without access to massive compute. Consequently, there is a pressing need to design compute-efficient SSL training method that still deliver robustness guarantees.

To address this gap, we propose a two-stage curriculum for DINOv2 pre-training that both speeds up convergence and enhances robustness to frequency-based corruptions—all without resorting to prohibitively large models or datasets. Our approach is motivated by the observation that high-frequency and low-frequency corruptions pollute different spectral bands of images, and that data-driven curricula can steer learning to emphasize or de-emphasize particular frequency bands at different training phases. The training curriculum consists of two stages:

**Stage 1 – Low-frequency training.** We begin by downsampling images, emphasizing their low-frequency components. This encourages the model to quickly learn broad, coarse features and accelerates convergence on clean data.

**Stage 2 – Full-resolution with high-frequency augmentation.** We then transition to full-resolution inputs while introducing Gaussian-noise patching, where random image patches are replaced with noise. This forces the model to learn invariances to high-frequency perturbations and improves robustness.

With extensive experiments, our method achieves not only faster convergence, but also higher robustness on models and datasets with various scales. This work demonstrates that robustness need not be an emergent by-product of extreme scale but can instead be built into SSL through careful curriculum design and augmentation. We believe this opens the door to more accessible, reproducible, and robust self-supervised training. In summary, our contributions are the following:

- We conduct a comprehensive robustness and frequency-based analysis on models pretrained with a low-frequency data curriculum—an underexplored direction. We identify the low-frequency bias introduced by this training scheme and propose Gaussian noise patching as a complementary augmentation to enhance robustness.

- The proposed curriculum accelerates convergence on ImageNet-1K [27] with DINOv2 and a ViT-B backbone, reducing pretraining time by $1.66\times$ and FLOPs by $2.25\times$, while maintaining matching robustness and competitive clean linear probing accuracy.

## 2    Related Work

**Frequency-Guided Curriculum Learning**    By ordering the example from the easiest to the hardest, curriculum learning yields a monotonic increase in the rate of convergence under stochastic gradient descent [34]. It has also been proven that an ideal curriculum can effectively smooth the optimization landscape without changing the location of global minima [14]. Thus, defining the level of difficulty effectively is essential for a powerful curriculum. Gradient-based training of deep networks systematically fits the low-frequency components of a target function first, and gradually capture high-frequency features as training proceeds [36]. For vision transformer, specifically, the multi-head self-attention acts as a low-pass filter, attenuating high-frequency signals while preserving low-frequency components [23].

The coarse-to-fine paradigm in computer vision motivates training models first in images with reduced information, then on their full-resolution counterparts. Therefore, it is natural to define low-frequency components from an image as easy examples, while full image with both low- and high-frequency features as more difficult ones. From a Fourier perspective, natural image content is concentrated in the low-frequency domain [37], so downsampling preserves most of the information while reducing computational cost. Several works incorporate small images or low-frequency components into the training pipeline to accelerate ViT convergence: RECLIP[19] applies this curriculum to pre-train CLIP, and EfficientTrain++ [33] generalizes it for various ViT-based models, including MoCo [15] and MAE [16]. However, the impact of this approach on model robustness remains unclear. Moreover, as a general method, EfficientTrain++ evaluated it on DINO [7], but did not observe convergence speedup. In this paper, we revisit the curriculum learning applied to DINOv2. Besides the acceleration in training convergence, we also discover that this data curriculum unexpectedly biases model toward high-frequency information.

**Frequency Bias to Robustness-Accuracy Tradeoffs**    Robustness research often adopts a frequency-domain perspective to examine how image corruptions correspond to the frequency spectrum [37]. In natural images, low-frequency components dominate, carrying most structural information, so high-frequency corruptions, such as Gaussian or shot noise, primarily disrupt fine details like edges, while low-frequency corruptions, including contrast shifts, brightness changes, or fog, modify broader patterns. Recent works have studied the need to model the noise as part of the data augmentation strategies–justifying the use of such corruptions as means to improve downstream robustness [31]. Different data augmentations introduce frequency biases that enhance robustness to specific corruptions but reveal distinct strengths and weaknesses [8]. For example, Gaussian noise augmentation strengthens robustness to high-frequency perturbations, producing a low-frequency-biased model, but often degrades performance on medium-frequency distortions like motion blur or Gaussian blur [37]. Similarly, CutOut [10] augmentation, which masks image regions to emphasize high-frequency cues such as edges or textures, encourages models to rely on features that are easily corrupted by noise or blur, limiting robustness gains. Adversarial training, on the other hand, biases models toward low-frequency features, enhancing resilience to certain perturbations but compromising performance on low-frequency corruptions [38] [6]. While these augmentations offer specific strengths, they often impair generalization by prioritizing certain frequency bands over others. In this work, we address this tradeoff by integrating Gaussian noise patching [20], which applies noise to localized image patches to balance robustness to high-frequency corruptions with preserved accuracy on clean images. This approach complements the high-frequency bias induced by our data curriculum, fostering a more balanced model across the frequency spectrum, though careful evaluation is needed to mitigate potential weaknesses on medium-frequency corruptions.

## 3    FastDINOv2: Recipe for Fast and Robust Pretraining

This section presents the proposed curriculum learning pipeline. At a high level, the pipeline incrementally introduces high-frequency features through a structured learning curriculum. Additionally, Gaussian noise patching is incorporated into the training process to enhance robustness.

### 3.1    Low-Frequency Feature Extraction

Low-frequency features can be extracted through multiple approaches. While methods like Efficient-Train++ [33] employ Fourier transform with high-frequency filtering for this purpose, we opt for

a simpler and more computationally efficient strategy. In our pipeline, we use downsampling as a lightweight proxy for low-frequency extraction, applied after DINOv2's standard image cropping preprocessing. In DINOv2's teacher-student framework, the teacher network processes global crops, which preserve low-frequency structural information, while the student network handles local crops, which retain high-frequency details. The cropping operation is defined as follows:

$$s \sim \mathcal{U}\big(s_{\min}, s_{\max}\big), \quad \tilde{w} = \tilde{h} = \sqrt{s\,H\,W} \tag{1}$$

where $s_{\min}(>= 0.08)$ and $s_{\max}(<= 1.0)$ define the crop's area ratio relative to the original image dimensions $(H, W)$. We adopt DINOv2's original scaling parameters: global crops use $(s_{\min}, s_{\max}) = (0.32, 1.0)$, while local crops use $(0.05, 0.32)$.

After cropping, downsampling is performed to extract low-frequency features. This operation uses bicubic interpolation, a resampling method that computes a weighted average of the nearest $4 \times 4$ neighborhood of input pixels through a cubic convolution kernel. The specific definition for cubic kernel function is in A.3. We reduce global crop sizes from $224 \times 224$ to $112 \times 112$ and local crops from $96 \times 96$ to $48 \times 48$.

## 3.2 Frequency Based Curriculum Learning

**Training Curriculum** Curriculum learning [5] improves model performance by progressively training on simpler samples before advancing to more complex ones. In our approach, we define low-frequency components of images—representing coarse, large-scale patterns—as the easier samples, gradually progressing to harder examples represented by the original images. Our curriculum consists of two stages. For the first 75% of training epochs, the model is trained exclusively on these low-frequency components. Subsequently, we apply a restarting mechanism by resetting the Adam optimizer's training dynamics. In the second stage, we train DINOv2 on original images containing both low- and high-frequency information for the remaining epochs. To ensure training stability, we maintain a fixed batch size across both stages.

**Balancing Frequency Bias via Gaussian Noise Patching** As previously mentioned, training with the low-frequency curriculum biases the model toward high-frequency features, enhancing robustness against low-frequency corruption compared to the standard DINOv2 training approach. To balance this bias and improve robustness to low-frequency features, we introduce Gaussian noise patching in the second stage of training. Combined with cut-out augmentation and Gaussian noise augmentation, this technique effectively encourages the model to focus on low-frequency features while maintaining performance on clean data. For implementation, we randomly select a square patch for each image, with side length $\tilde{h} = \tilde{w} < \min(H, W)$, where $H$ and $W$ are the image's height and width before transformation. We then apply Gaussian noise to this patch, perturbing each pixel value $x$ with a noise value $\tilde{x}$ independently sampled from a normal distribution $\tilde{x} \sim \mathcal{N}(1, \text{scale}^2)$. The noise intensity is controlled by the parameter scale.

## 4 Experiments

### 4.1 Datasets and Training Setup

**Datasets** For faster experimentation, we primarily use ImageNet-100 [32], a subset of ImageNet-1K [27]. The training set consists of 100 randomly selected classes from ImageNet-1K, with the first 500 images from each class. Similarly, the validation set contains the corresponding 100 classes from the original validation set, with 50 images per class. This results in a total of 50,000 training images and 5,000 validation images. For robustness evaluation, we use ImageNet-100-C, derived from ImageNet-C [17], which benchmarks model resilience to common corruptions. We maintain the exact image selection from the ImageNet-100 validation set across all corruption levels and types. Additionally, we employ ADE20K for semantic segmentation tasks. Finally, we scale our approach to full ImageNet-1K and evaluate robustness on ImageNet-C.

**Model and Training Details** For ImageNet-100 experiments, we use ViT-S/16 as the DINOv2 backbone with a total batch size of 40, distributed across 4 GPUs (10 per GPU). Experiments for large batch size in first stage of training are in Appendix A.1. We adopt positional embedding with

interpolation for adapting to image resolution shift between two training stages. The results of applying positional embedding with or without interpolation across different embedding sizes are in Appendix A.2. We train baseline models for 500 epochs and training curriculum experiments for 200 epochs, ensuring the baseline converges to optimal performance. All ImageNet-100 experiments use a fixed epoch length of 1,250 iterations. For ImageNet-1K experiments, we employ ViT-B/16 with a total batch size of 512 (128 per GPU), with epoch length of 2,500 iterations. The baseline and FastDINOv2 are trained for 250 and 200 epochs on ImageNet-1K, respectively. Following the official DINOv2 implementation, we use AdamW optimizer with square root learning rate scaling based on batch size, yielding a base learning rate of $7.9 \times 10^{-4}$. For linear probing evaluation, we use a batch size of 128 with 12.5k total iterations. All training runs are distributed across 4 NVIDIA L40S GPUs, while evaluations use either NVIDIA A6000 or NVIDIA A5500 GPUs.

## 4.2 Linear Probing Reveals Fast Convergence Without Compromising Accuracy

We evaluate our low-frequency data curriculum against the DINOv2 baseline through linear probing, training linear classifiers on frozen backbones. The baseline ViT-S/16 model follows standard DINOv2 training for 500 epochs with consistent $224 \times 224$ resolution. Our curriculum employs $112 \times 112$ images for the first 75% of training (150 epochs), then transitions to $224 \times 224$ images for the remaining epochs (denoted as 112-224 curriculum).

Notably, the 112-224 curriculum achieves the baseline's 250-epoch accuracy by epoch 200, demonstrating 20% faster convergence in training epochs and $1.44 \times$ acceleration in training time. We additionally validate the framework's flexibility by testing alternative first-stage resolutions in Table 2, observing consistent efficiency gains across configurations.

The $112 \times 112$ resolution in our curriculum's initial phase reduces input tokens per forward pass by 75% compared to baseline (Table 2), yielding substantial computational savings. However, we identify a critical tradeoff: excessively small first-stage resolutions (e.g., $64 \times 64$ or $96 \times 96$) degrade linear probing performance due to insufficient learning signal. While these ultra-low resolutions marginally improve training speed, they exhibit diminishing returns in efficiency gains compared to $112 \times 112$. Thus, we select $112 \times 112$ as the optimal first-stage resolution - balancing computational efficiency with preserved information.

Table 2: Linear probing accuracy on ImageNet-100. Our data curriculum pipeline matches DINOv2 performance while achieving a $1.44 \times$ convergence speed-up. 112-224 means that the FastDINOv2 training utilizes the 112-224 data curriculum; same applies to the remaining training methods. The training time is measured in NVIDIA L40S hours on a single GPU.

| Training method | Accuracy | Training epoch | Training time |
|---|---|---|---|
| DINOv2 | 78.6% | 250 | 24h |
| **112-224 FastDINOv2 w/o GP** | **78.44%** | 200 | 13.9h |
| 128-224 FastDINOv2 w/o GP | 77.74% | 200 | 13.6h |
| 96-224 FastDINOv2 w/o GP | 77.2% | 200 | 12.9h |
| 64-224 FastDINOv2 w/o GP | 70.6% | 200 | 13.48h |

## 4.3 Data Curriculum Induces High-Frequency Feature Bias

While previous sections analyzed the curriculum design and its training acceleration benefits, we now investigate how low-frequency data curriculum impacts model robustness against common corruptions. Using ImageNet-C as our corruption robustness benchmark, we construct ImageNet-100-C by preserving corresponding classes and images from the ImageNet-100 validation set.

Our evaluation protocol trains linear classifiers exclusively on clean ImageNet-100 validation data using frozen backbones, then tests them on the corrupted ImageNet-100-C images. This ensures neither the model nor classifier encounters corrupted data during training. We compute average error rates across all corruption types and severity levels to quantify robustness.

In Table 3, our curriculum-trained models exhibit greater robustness to low-frequency corruptions compared to baseline, indicating stronger reliance on high-frequency features for classification. This emerges because high-frequency exposure during later training epochs forces the model to develop invariant representations of these components, thereby amplifying sensitivity to high-frequency

corruptions. The results confirm that our curriculum induces a high-frequency bias – an interesting product of the training strategy.

To better interpret this shift in robustness, we group corruption types in Table 3 based on their dominant frequency characteristics, following a Fourier analysis approach [37]:

- **Low-frequency corruptions:** Brightness, contrast, fog, frost
- **Mid-frequency perturbations:** Motion blur, defocus blur, glass blur, Gaussian blur, snow, zoom blur
- **High-frequency distortions:** Gaussian noise, impulse noise, shot noise, speckle noise
- **Hybrid effects:** Elastic transform, JPEG compression, pixelate, saturate, spatter

Table 3: ImageNet-100-C testing accuracies for DINOv2 baseline trained with 250 epochs, 112–224 data curriculum trained with 200 epochs, DINOv2 baseline trained with 200 epochs, and DINOv2 with Gaussian noise patching trained by 200 epochs. Comparison between DINOv2(250Ep) and 112-224(200Ep) suggests a low-frequency bias induced by data curriculum, while the comparison between DINOv2(200Ep) and DINOv2 w/ GP reveals the high-frequency bias from Gaussian noise patching.

| Corruption type | Test 112–224 Curriculum | | Test Gaussian Patching(200 Ep) | |
| --- | --- | --- | --- | --- |
| | DINOv2(250Ep) | 112-224 (200Ep) | DINOv2 | DINOv2 w/ GP |
| Brightness | 71.23% | **71.60%** | 64.76% | **66.43%** |
| Contrast | 51.49% | **55.24%** | 45.83% | **47.75%** |
| Fog | 47.06% | **48.22%** | 38.42% | **40.18%** |
| Frost | 43.80% | **45.45%** | 36.58% | **39.95%** |
| Defocus blur | **42.01%** | 40.24% | **33.80%** | 32.69% |
| Gaussian blur | **44.38%** | 41.94% | **37.15%** | 35.36% |
| Glass blur | 36.85% | **36.94%** | 29.36% | **30.56%** |
| Motion blur | 43.64% | **45.88%** | 37.37% | **38.21%** |
| Snow | 37.25% | **38.08%** | **30.22%** | 28.84% |
| Zoom blur | 47.26% | **47.40%** | **42.12%** | 41.54% |
| Gaussian noise | **32.11%** | 29.59% | 21.72% | **48.72%** |
| Impulse noise | **26.97%** | 25.62% | 17.34% | **44.89%** |
| Shot noise | **31.06%** | 29.04% | 21.30% | **45.68%** |
| Speckle noise | **40.87%** | 39.30% | 31.25% | **50.46%** |
| Elastic transform | **62.92%** | 62.07% | 56.70% | **57.17%** |
| JPEG compression | **58.58%** | 58.02% | 51.71% | **52.44%** |
| Pixelate | **51.79%** | 50.46% | **44.98%** | 44.47% |
| Saturate | **65.59%** | 64.89% | 57.39% | **58.74%** |
| Spatter | 55.12% | **55.34%** | 48.58% | **49.60%** |
| Corruption accuracy | **46.84%** | 46.60% | 39.29% | **44.93%** |
| Clean accuracy | **78.60%** | 78.42% | 73.46% | **74.05%** |

## 4.4 Spectral Balance: Combining Curriculum and Noise Patching

Building on the high-frequency feature bias observed in the training curriculum, we demonstrate how Gaussian noise patching, a low-frequency-biased augmentation, can counterbalance this effect. Importantly, integrating these approaches retains their individual strengths while addressing their respective limitations.

We identify Gaussian noise augmentation [1] as a method to introduce low-frequency bias, demonstrated by enhanced robustness against high-frequency noise corruptions. However, its direct application decreases clean accuracy due to excessive low-frequency emphasis. Gaussian noise patching mitigates this by selectively applying localized noise injection alongside preserved clean regions, thereby maintaining discriminative feature learning.

Table 3 compares the DINOv2 baseline with DINOv2 trained using Gaussian noise patching. The patched model exhibits a low-frequency bias, demonstrating greater robustness against high-frequency corruptions, including all noise corruptions. This frequency bias directly opposes that of our data curriculum, prompting the question: can Gaussian noise patching be integrated to balance the curriculum's frequency bias?

When applied solely during the high-resolution phase of the curriculum, this hybrid approach achieves balanced robustness improvements. As shown in Table 4, most corruption types show enhanced resilience, with minor reductions confined to specific mid-frequency distortions (e.g., zoom blur, elastic transform) and high-frequency artifacts (e.g., pixelate). Notably, the combined method eliminates conflicting vulnerabilities observed in individual applications, significantly improving robustness to defocus blur and Gaussian blur compared to either technique alone.

The synergy stems from complementary frequency interactions: the curriculum's early low-resolution training builds robust low-frequency representations, while subsequent Gaussian noise patching mitigates overfitting to artificial high-frequency patterns. This integration achieves a spectral balance, ensuring neither frequency domain overly dominates feature encoding.

These findings underscore the value of aligning augmentation strategies with curriculum stages. Although trade-offs remain for certain corruption types, this framework offers a pathway to balance spectral biases while maintaining core model performance.

Table 4: ImageNet-100-C test accuracy for DINOv2 baseline versus DINOv2 with data curriculum and Gaussian noise patching. Accuracy improves for most corruption types, demonstrating the effectiveness of combining data curriculum and Gaussian noise patching.

| Corruption type | DINOv2 | FastDINOv2 | Δ |
|---|---|---|---|
| Brightness | 71.23% | **71.92%** | +0.69% |
| Contrast | 51.49% | **56.09%** | +4.60% |
| Fog | 47.06% | **47.90%** | +0.84% |
| Frost | 43.80% | **47.52%** | +3.72% |
| Defocus blur | 42.01% | **42.31%** | +0.30% |
| Gaussian blur | 44.38% | **44.44%** | +0.06% |
| Glass blur | 36.85% | **40.24%** | +3.39% |
| Motion blur | 43.64% | **45.43%** | +1.79% |
| Snow | 37.25% | **37.96%** | +0.71% |
| Zoom blur | **47.26%** | 47.00% | −0.26% |
| Gaussian noise | 32.11% | **57.51%** | +25.40% |
| Impulse noise | 26.97% | **54.62%** | +27.65% |
| Shot noise | 31.06% | **55.34%** | +24.28% |
| Speckle noise | 40.87% | **61.59%** | +20.72% |
| Elastic transform | **62.92%** | 62.36% | −0.56% |
| JPEG compression | 58.58% | **61.15%** | +2.57% |
| Pixelate | **51.79%** | 49.18% | −2.61% |
| Saturate | 65.59% | **65.74%** | +0.15% |
| Spatter | 55.12% | **56.35%** | +1.23% |
| Corruption accuracy | 46.84% | **52.88%** | +6.04% |
| Clean accuracy | **78.6%** | 78.4% | −0.20% |

## 4.5 Instance Recognition Indicates Instance-Level Effectiveness

Building on the demonstrated performance in linear probing, we further evaluate FastDINOv2 on an instance-level task through instance recognition. Specifically, we assess three difficulty levels on the Oxford and Paris datasets and report the mean average precision (mAP) in table 5. On the Oxford dataset, FastDINOv2 outperforms DINOv2 by 3.73% on the easy split and 1.22% on the medium split, with a slight drop on the hard split. Both models exhibit comparable performance on the Paris dataset. These results indicate that FastDINOv2 not only accelerates convergence but also enhances performance across a variety of tasks.

Table 5: Mean Average Precision (mAP) on instance-level recognition tasks using Oxford and Paris datasets (E: Easy, M: Medium, H: Hard)

| Training method | Oxford | | | Paris | | |
|---|---|---|---|---|---|---|
| | E | M | H | E | M | H |
| FastDINOv2 | 32.11% | 22.26% | 3.99% | 56.38% | 40.71% | 14.56% |
| DINOv2 | 28.38% | 21.04% | 4.56% | 56.88% | 40.76% | 14.94% |

## 4.6 Semantic Segmentation Suggests Intact Pixel-Level Performance

Section 4.2 and 4.5 show that the FastDINOv2 achieves faster convergence while maintaining linear probing and improving instance recognition accuracy, proving that image and instance-level task remains strong. However, since semantic segmentation requires fine-grained pixel understanding, we needed to verify whether our approach - which spends 75% of training time on low-frequency features - could still perform well on such tasks. To test this, we conducted semantic segmentation experiments using ADE20K. We froze the DINOv2 backbone (trained with our curriculum) and trained only a 2-layer convolutional decoder, evaluating performance using both mean Intersection-over-Union (mIoU) and mean class accuracy (mAcc) metrics. Table 6 demonstrates that our 112-224 curriculum with Gaussian patching matches the DINOv2 baseline's segmentation performance (mIoU/mAcc) in just 200 epochs. Despite initial low-frequency training, the second stage successfully recovers fine-grained pixel understanding needed for segmentation tasks.

Table 6: Semantic segmentation evaluation with frozen DINOv2 backbone. A 2-layer convolutional decoder is trained on top of frozen DINOv2 using ADE20K dataset for 200 epochs.

| Training Method | Pre-training Epochs | Clean Accuracy | mAcc | mIoU |
|---|---|---|---|---|
| DINOv2 | 250 | 78.6% | 25.09% | 19.2 |
| FastDINOv2 | 200 | 78.4% | 25.21% | 19.16 |

## 4.7 Frequency Based Analysis and Grad-CAM Visual Explanation

In addition to the evaluations using ImageNet-100-C and linear probing, we explore whether different training paradigms influence the error sensitivity of models across all frequency bands—not only those in ImageNet-100-C—and the regions the model prioritizes during predictions. To this end, we generate Grad-CAM visualizations [29] and Fourier spectral heatmaps following the methods in [37].

Figure 2 shows the error sensitivity of the resulting model across various frequency ranges. Compared to the DINOv2 baseline, our approach achieves lower error rates in high- and medium-frequency bands. In low-to-mid frequency ranges, particularly in the central region, our method reduces the error rate by a significant margin. However, the model shows increased susceptibility to perturbations in the mid-to-high frequency range, indicating potential trade-offs in robustness across frequency bands.

Figure 3 presents Grad-CAM visualizations that highlight further differences. While the DINOv2 baseline distributes its focus across both the target object and background areas during predictions, the model trained with FastDINOv2 emphasizes object contours more strongly. This suggests that the initial stage of pre-training provides an effective initialization, enabling the model to focus on coarse image components.

## 4.8 Extending to ViT-B and ImageNet-1K: Balancing Efficiency and Robustness

To validate scalability, we extend our framework to ViT-B architectures and ImageNet-1K pretraining. The baseline DINOv2 model is pretrained for 200 epochs using standard protocols, while our FastDINOv2 method combines a 112-224 curriculum with Gaussian noise patching over an equivalent number of epochs—allocating 150 epochs to downsampled inputs followed by 50 epochs of full-resolution training. Evaluation includes linear probing on clean ImageNet-1K validation data and robustness assessment using ImageNet-C with frozen backbones.

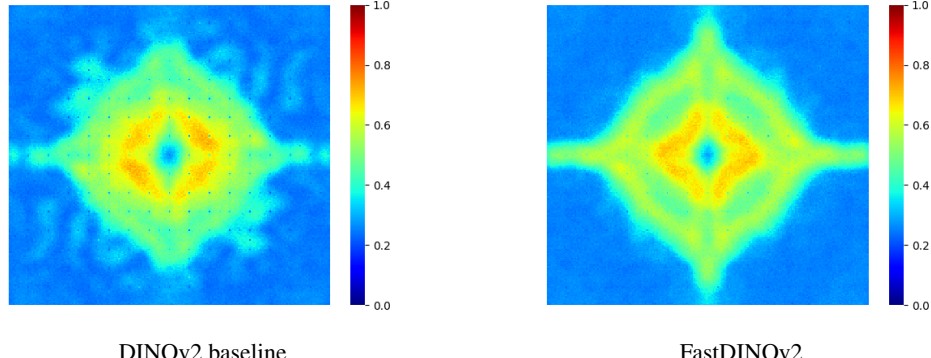

DINOv2 baseline                                           FastDINOv2

Figure 2: Fourier error sensitivity heatmap of model trained with our method and DINOv2 baseline. The heatmap is generated with a subset of Imagenet-100 validation set with 5 images sampled from each class. Color indicates the error sensitivity to that specific frequency range. Low-frequency features are mainly concentrated into center area, while the region further away from center represents feature with high frequency.

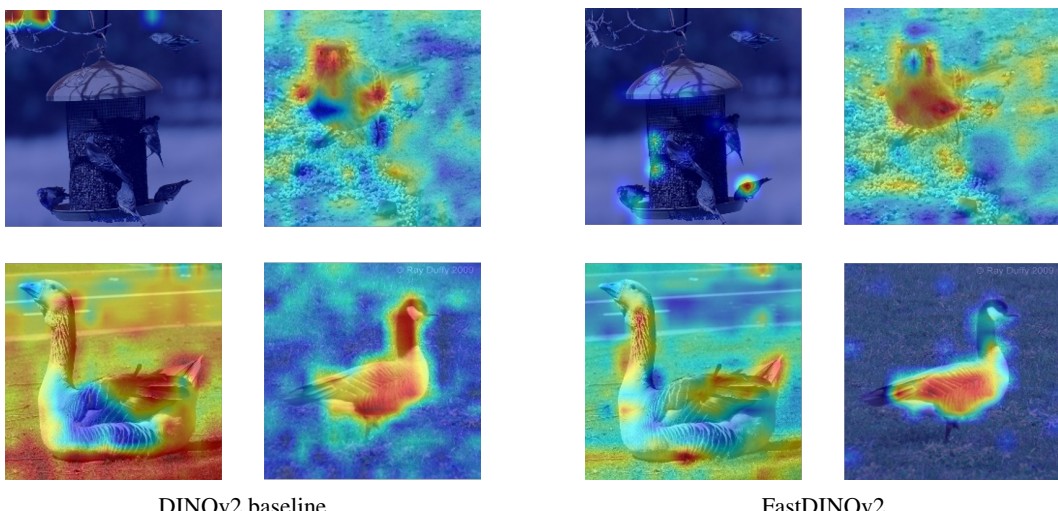

DINOv2 baseline                                           FastDINOv2

Figure 3: Grad-CAM maps for DINOv2 baseline and FastDINOv2. The first row images are from class "house finch, linnet, Carpodacus mexicanus", and the second row from "goose". With the data curriculum, model can better capture the contour of the object. See more examples in A.7

Table 1 reports the evaluation results and training costs for the baseline and our method. Although training time is reduced by $1.6\times$ and FLOPs by $2.25\times$, our method achieves competitive linear probing accuracy. Moreover, testing on ImageNet-C with a frozen backbone and linear classifier shows a comparable error rate despite a slight drop in clean accuracy, demonstrating the effectiveness of our method in enhancing robustness.

Huge memory consumption for GPU is a critical bottleneck for scaling up the pretraining. However, with low-resolution image in the first training stage, FastDINOv2 allows a much lower memory requirement. As in table 7, during the first 75% of training epochs the maximum memory consumption of FastDINOv2 is 9.47GB for batch size of 128 per GPU, significantly less than 33.5GB for DINOv2 baseline with same batch size. This reduction in memory requirement shows possibilities for a cost-effective model by leveraging GPUs with lower memory capacity for most of the pretraining epochs.

Table 7: Comparison of training epochs and max memory consumptions on a single NVIDIA L40S GPU with 48GB memory. Both DINOv2 and FastDINOv2 apply ViT-B backbone trained on ImageNet-1K.

| Training Method | Training Epochs | | Max Memory Consumption(GB) | |
|---|---|---|---|---|
| | Low-Res | Full-Res | Low-Res | Full-Res |
| DINOv2 | - | 200 | - | 33.5 |
| **FastDINOv2** | 150 | 50 | 9.47 | 33.5 |

## 5 Limitations, Future Work, and Conclusion

In this work, we presented an efficient training framework for the vision foundation model DINOv2, demonstrating its scalability and robustness across diverse datasets and model backbones. Our method achieves substantial improvements in training efficiency, computational cost reduction, and model performance. However, one limitation of our current approach lies in the fixed schedule for transitioning from low-resolution to standard-resolution images during the final 25% of training epochs. This heuristic, while effective in our experiments, may not be optimal across different hyperparameter settings or training regimes. Future work will focus on developing adaptive scheduling strategies that dynamically determine the transition point based on training signals such as convergence metrics or validation performance. Additionally, we plan to investigate how our resolution-aware curriculum can be combined with other training enhancements such as adversarial training paradigms.

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

# A Technical Appendices and Supplementary Material

## A.1 Constant Batch Size Ensures Training Stability

Smaller image sizes consume less GPU memory during training, which naturally suggests using a larger batch size in the first stage of the curriculum. To investigate the effect of an early large batch size, we test the 112–224 curriculum by increasing the batch size from 40 to 60 in the first stage while keeping the second stage unchanged. The base learning rate for the first stage is also scaled proportionally from $4 \times 10^{-3}$ to $6 \times 10^{-3}$. As shown in Table 8, while the training time is slightly reduced with the early large batch size and scaled learning rate, the clean accuracy decreases. This suggests that varying the batch size during training can introduce instability and degrade model performance. Therefore, we use a constant batch size in all subsequent experiments.

Table 8: Linear probing accuracy for the early large batch size and learning rate. Notice that in this experiment, FastDINOv2 training utilizes 112-224 data curriculum and does not apply the interpolation for positional embedding and Gaussian patching in order to exclude their effects. In the Batch size column, $60 + 40$ indicates that batch size of 60 is used in the first stage, and 40 for second stage. Same for learning rate. Training cost is measured in NVIDIA L40S hours for a single GPU.

| Training method | Accuracy | Training cost(hours) | Batch size | Learning rate |
|---|---|---|---|---|
| DINOv2 | 78.6% | 250 epochs, 24 | 40 | $4e-3$ |
| FastDINOv2 w/o GP | 78.06% | 200 epochs, 12.3 | $60 + 40$ | $6e-3 + 4e-3$ |
| FastDINOv2 w/o GP | 78.42% | 200 epochs, 13.9 | 40 | $4e-3$ |

## A.2 Positional Embedding for Varying Image Resolution

DINOv2 uses learnable positional embeddings, which capture the location information of input patches during training. In the first half of the training curriculum, images are downsampled to $112 \times 112$ as input, while in the second stage, regular-sized images ($224 \times 224$) are used. This setup requires the model to adapt to multiple input resolutions. Two approaches are available: either using separate positional embeddings for each stage or using a consistent positional embedding and interpolating it. The second approach introduces two further choices: interpolating the positional embedding based on either the first stage's smaller resolution ($112 \times 112$) or the second stage's full resolution ($224 \times 224$). In Table 9, we compare these settings and find that interpolation based on the first stage's smaller resolution ($112 \times 112$) performs best. After the first training stage, we retain the positional embedding and apply bicubic interpolation for upsampling, adjusting it to the higher resolution in the second stage. This process mirrors our image downsampling procedure, where we also use a bicubic kernel to resample pixels.

Table 9: Linear probing accuracy on Imagenet-100 for FastDINOv2 with 112-224 data curriculum to compare the following three design: interpolation with fixed $112 \times 112$ positional embedding, interpolation with fixed $224 \times 224$ positional embedding, and using separate positional embeddings(no reusing). In the third design option, positional embedding from the first stage is abandoned, while a new positional embedding is trained based on second-stage's image resolution. FastDINOv2 training in this experiment does not use Gaussian patching to eliminate its effect.

| Training method | Accuracy | Training epoch | Positional embedding |
|---|---|---|---|
| DINOv2 | 78.6% | 250 | no interpolation |
| **FastDINOv2 w/o GP** | **78.8%** | 200 | interpolation w/ $112 \times 112$ |
| FastDINOv2 w/o GP | 77% | 200 | interpolation w/ $224 \times 224$ |
| FastDINOv2 w/o GP | 78.42% | 200 | no reusing |

## A.3 Down-sampling with Bicubic Interpolation

We adopt Bicubic interpolation for down-sampling to obtain low-frequency inputs for first-stage training. The cubic kernel $w(t)$ is defined as:

$$w(t) = \begin{cases} (a+2)\,|t|^3 \; - \; (a+3)\,|t|^2 \; + \; 1, & |t| \le 1, \\ a\,|t|^3 \; - \; 5a\,|t|^2 \; + \; 8a\,|t| \; - \; 4a, & 1 < |t| < 2, \quad (a = -0.5) \\ 0, & |t| \ge 2, \end{cases} \tag{2}$$

where $t$ represents the normalized distance from the sampling point to a neighboring pixel, and $a$ controls the curvature of the cubic kernel. Using this kernel function, the bicubic interpolation $f(x,y)$ is computed by applying the convolution kernel $w$ separably along both spatial dimensions:

$$f(x,y) = \sum_{m=-1}^{2} \sum_{n=-1}^{2} w\big(m - \Delta x\big)\, w\big(n - \Delta y\big)\, f\big(x_0 + m,\ y_0 + n\big); \quad m, n \in \{-1, 0, 1, 2\} \tag{3}$$

For an input image of size $(H_{\text{orig}}, W_{\text{orig}})$ and target downsampled size $(H_{\text{down}}, W_{\text{down}})$, the coordinate mapping transforms each pixel location $(u, v)$ in the original image to $(x, y)$ in the downsampled space as:

$$(x, y) = (u\,\frac{W_{\text{orig}}}{W_{\text{down}}}, v\,\frac{H_{\text{orig}}}{H_{\text{down}}}), \quad (x_0, y_0) = (\lfloor x \rfloor, \lfloor y \rfloor), \quad (\Delta x, \Delta y) = (x - x_0, y - y_0) \tag{4}$$

### A.4    Experiments on SimCLR

In order to examine the generalization ability of our method across frameworks and model architectures, we conducted additional experiments by training a SimCLR model with ResNet backbone using our FastDINOv2 method (denoted as "FastSimCLR"). Using the LARS optimizer, the SimCLR baseline was trained for 200 epochs on full-resolution images from ImageNet-100, while FastSimCLR was trained for 150 epochs on low-frequency data followed by 50 epochs on full-resolution images. Despite the differences in inductive biases between ViTs and CNNs, in table 10 FastSimCLR shows faster convergence. Further, combining this curriculum with Gaussian noise patching improves robustness toward Imagenet-C.

Table 10: Linear probing accuracy on ImageNet-100 and testing accuracy on ImageNet-100-C for SimCLR and FastSimCLR. Training time is measured in NVIDIA A6000 hours for a single GPU.

| Training Method | Epochs | ImageNet-100 ($\uparrow$) | ImageNet-100-C ($\uparrow$) | Training Time |
|---|---|---|---|---|
| SimCLR | 200 | 64.48% | 30.95% | 14 |
| FastSimCLR | 150+50 | 64.82% | 38.73% | 11.2 |

### A.5    Ablation Experiment for Transition Epoch

One critical design for FastDINOv2 pre-training framework is the transitioning epochs, where training data shifts from low-resolution images to full-resolution ones. Correctly choosing this transition point is essential - if the training epoch for the first stage is insufficient, the model fails to take advantage from low-resolution images to improve training efficiency; on the other hand, the model lacks exposure to the full-resolution features necessary for downstream tasks if the first stage is excessive. Thus, we determine this transition point empirically by conducting a set of cross validation experiments with different transition epoch in table 11.

Among all the splits, the 75/25 splits consistently yielded the best trade-off, achieving the highest linear probing accuracy. More extreme cases such as 0/100 would have no aceleration effect, and for 100/0 the model would only be trained on low-frequency data, which lead to notably worse performance.

Table 11: Linear probing accuracy on ImageNet-100 for FastDINOv2 with different transition epochs proportion. The number of total training epoch is 200, and training time is measured in NVIDIA A6000 hours for a single GPU.

| Training Method | Split (Stage1/Stage2) | Training Time | ImageNet-100 |
|---|---|---|---|
| FastDINOv2 | 65/35 | 20.4 | 77.6% |
| FastDINOv2 | 75/25 | 20.4 | 78.1% |
| FastDINOv2 | 85/15 | 20.68 | 76.8% |
| FastDINOv2 | 0/100 | 22.3 | 73.9% |
| FastDINOv2 | 100/0 | 20.1 | 69.7% |

## A.6 Convergence Speed Comparison

In order to better demonstrate the trend of convergence speed of FastDINOv2, we measure the training time gap for reaching several linear probing accuracy levels for both FastDINOv2 and DINOv2 baseline in table 12.

Table 12: Training time comparison (in NVIDIA A6000 hours) required to reach different accuracy levels on ImageNet-100.

| Training Method | 55% Accuracy | 65% Accuracy | 75% Accuracy | 78% Accuracy |
|---|---|---|---|---|
| DINOv2 | 2.6h | 4.8h | 12.48h | 23.04h |
| FastDINOv2 | 2.64h | 5.28h | 11.5h | 13.1h |

During the low-frequency training stage (when FastDINOv2 reaches 55% and 65% accuracy), FastDINOv2 achieves similar convergence speed to the baseline trained on full-resolution images. This is attributed to the inherent low-frequency bias in deep neural networks and transformers, such that the model is more inclined to learn low-frequency information first during training. Furthermore, this low-frequency stage provides a strong foundation for the subsequent full-resolution training stage (when FastDINOv2 reaches 75% and 78% accuracy), enabling FastDINOv2 to more efficiently and effectively learn fine-grained, high-resolution details from full-resolution images.

## A.7 GradCAM Visualization and Imagenet-C Examples

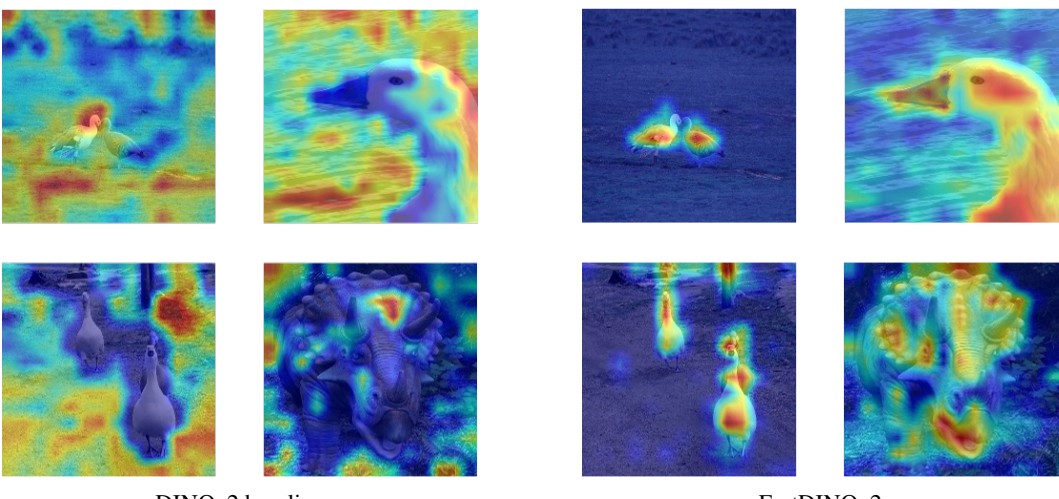

DINOv2 baseline                                   FastDINOv2

Figure 4: Extra Grad-CAM maps examples for DINOv2 baseline and FastDINOv2.

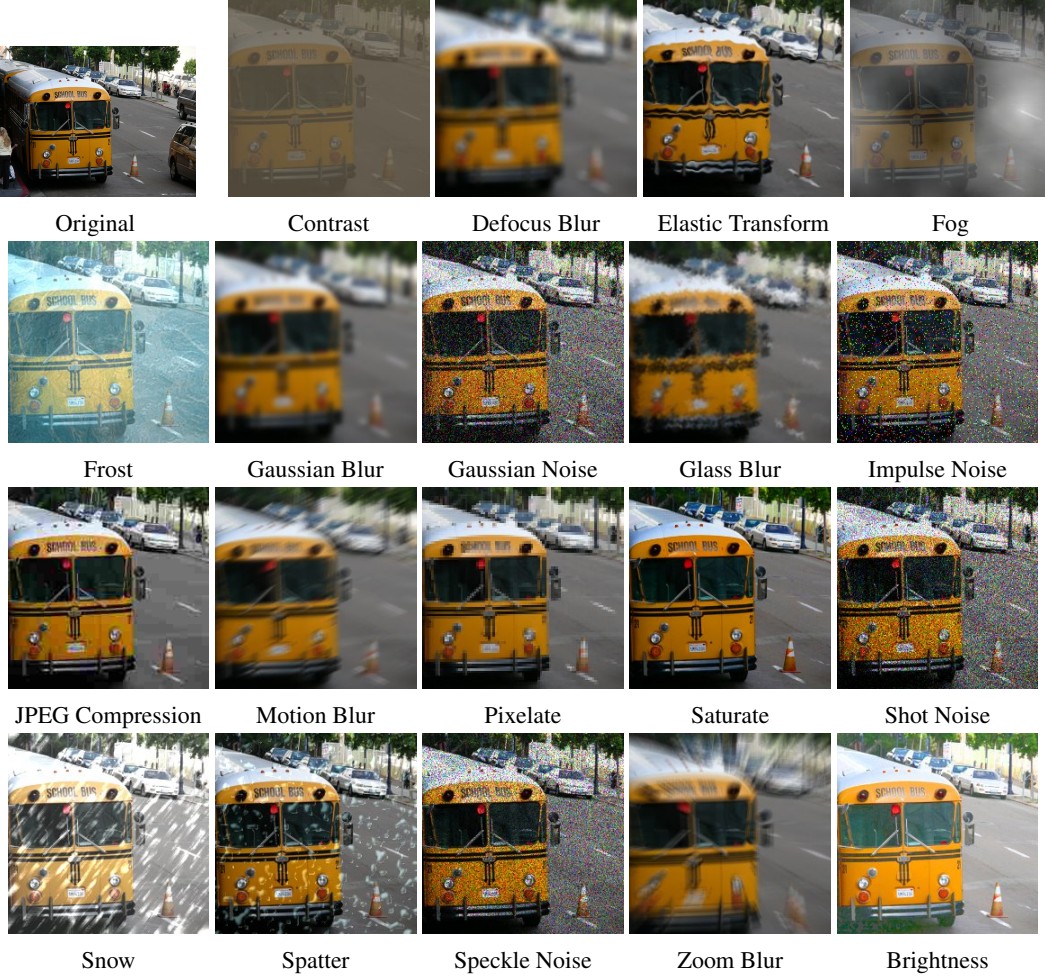

Figure 5: Imagenet-C examples for each corruption type.

