# OpenReview forum: "FastDINOv2: Frequency Based Curriculum Learning Improves Robustness and Training Speed"
_NeurIPS.cc/2025/Conference — NeurIPS 2025 poster_

### Official Review · Reviewer_keDD · 2025-06-18

**Clarity:** 3
**Significance:** 2
**Originality:** 2
**Rating:** 3
**Confidence:** 3

**Summary:**

This work proposes a novel pretraining strategy for DINOv2 to accelerate training process, meanwhile the strategy improves robustness of distortion. The approach is based on frequency-based curriculum learning to see low-frequency images in low-resolution first and then apply guassian patches to high-resolution images for high-frequency training. As a result, the pretraining time is reduced by 1.6 times for a ViT-B/16 on ImageNet. The probing results on ImageNet-C and ImageNet verified the benefits of model robustness and training speed from Frequency-Based Curriculum Learning.

**Questions:**

A similar strategy is popular in progressive training, which increases the resolution and the augmentation strength gradually during training [1,2]. As mentioned in [1], low-resolution does not mean an east task in MAE. In contrast, low-resolution images could act as regularisation to improve performance[3].

Why use fewer training epochs? To show the speedup number, the final performance should match the original number.



[1]: Wu J, Mo S, Atito S, et al. Dailymae: Towards pretraining masked autoencoders in one day[C]//European Conference on Computer Vision. Springer, Cham, 2025: 131-149.

[2]: Howard, J.: fast.ai - Now anyone can train Imagenet in 18 minutes (Aug 2018), https://www.fast.ai/posts/2018-08-10-fastai-diu-imagenet.html
[3]: Li X, Wang Z, Xie C. An inverse scaling law for clip training[J]. Advances in Neural Information Processing Systems, 2023, 36: 49068-49087.

**Ethical Concerns:**

["NO or VERY MINOR ethics concerns only"]

**Final Justification:**

Thank you for your updated results. I will raise my rating but won't be high. The contribution to accelerate training is good, but the weakness are also clear. First, there are some theories to explain such behaviours during training that the model prefers low-frequency components at the beginning stage [1,2]. Since the model cannot learn low-frequency components, we can just remove high-frequency components to speed up the training, which has been applied in many studies. The paper lacks a comprehensive discussion about these theories and work. Second,  the real question is how to effectively extract high-frequency components. Although this paper proposed an easy yet effective solution by Gaussian Patching, it lacks enough experimental results to demonstrate its effectiveness. Third,  the Gaussian Patching does not sound an ultimate solution to enforce models to learning high-frequency components. In fact, there is relevant study on this [4].


[1] Balestriero R, LeCun Y. Learning by reconstruction produces uninformative features for perception[J]. arXiv preprint arXiv:2402.11337, 2024.

[2] Littwin E, Saremi O, Advani M, et al. How jepa avoids noisy features: The implicit bias of deep linear self distillation networks[J]. arXiv preprint arXiv:2407.03475, 2024.

[3] Wang H, Wu X, Huang Z, et al. High-frequency component helps explain the generalization of convolutional neural networks[C]//Proceedings of the IEEE/CVF conference on computer vision and pattern recognition. 2020: 8684-8694.

[4] Bizeul A, Sutter T M, Ryser A, et al. COMPONENTS BEAT PATCHES: EIGENVECTOR MASKING FOR VISUAL REPRESENTATION LEARNING[J].

**Limitations:**

yes

**Quality:**

2

**Strengths And Weaknesses:**

Strength:
1. From Table 3 and 4, we can see clear improvement on robustness by introducing Gaussian noise patching.
2. The idea is simple and hopefully applied in SSL families.


Weakness:
1. The experimental results can hardly support their claims: The final performance of the 112-224 curriculum (78.44%) is lower than the original DINOv2 (78.6%) for  linear probing on IN100. A lower performance on dense tasks in Table 5.
2. The conclusion is made on ImageNet-100, an easy subset with less noise. For large-scale datasets full of diverse noise, the impact of Gaussian noise patching is questionable.
3. How to combine masked image modelling and Gaussian noise patching is not clear. Does the GP applied to visible patches only? Or a replacement of masking?
4. This work lacks a comprehensive literature review on the studies of frequency and robustness.

---

> ### Author Rebuttal · Authors · 2025-07-31
>
> We sincerely thank Reviewer keDD for the insightful questions. We would like to give detailed responses to each of your comments and questions.
>
> ### Question 1. The experimental results can hardly support their claims:The final performance of the 112-224 curriculum (78.44%) is lower than the original DINOv2 (78.6%) for linear probing on IN100. A lower performance on dense tasks in Table 5.
> We thank the reviewer for this important observation. We understand the reviewer’s concern regarding the final performance of the 112–224 curriculum. While performance improvement is valuable, the central contribution of our work lies in maintaining accuracy and robustness while significantly reducing training cost, and in providing a comprehensive analysis of frequency bias in the resulting model.
> With only a 0.2% drop in accuracy, the 112–224 curriculum achieves both a robustness gain and a substantial reduction in training time. As noted by the reviewer, the dense task results in Table 5 further support this: FastDINOv2 achieves a 0.21% improvement in mAcc with only a 0.04% drop in mIoU, indicating comparable performance across both image-level and pixel-level tasks.
> To further assess the effectiveness of our method on downstream applications, we also conducted additional experiments on instance-level recognition using the Oxford and Paris datasets, each evaluated with three difficulty-level splits. On the Paris dataset, FastDINOv2 achieves comparable performance across all levels. On the Oxford dataset, our model shows substantial improvements on the easy and medium splits, further demonstrating the robustness and generalization of our approach.
> | Training method | Oxford E | Oxford M | Oxford H | Paris E | Paris M | Paris H |
> |-----------------|----------|----------|----------|---------|---------|---------|
> | FastDINOv2      | 32.11%   | 22.26%   | 3.99%    | 56.38%  | 40.71%  | 14.56%  |
> | DINOv2          | 28.38%   | 21.04%   | 4.56%    | 56.88%  | 40.76%  | 14.94%  |
>
> ### Question 2. The conclusion is made on ImageNet-100, an easy subset with less noise. For large-scale datasets full of diverse noise, the impact of Gaussian noise patching is questionable.
> We thank the reviewer for raising this important point regarding the potential impact of training on larger and noisier datasets. In Table 1, we provide results for both linear probing and robustness evaluation on ImageNet-1K and ImageNet-C. These results show that FastDINOv2 significantly reduces training time, with only a slight drop in linear probing accuracy and comparable robustness—demonstrating that the proposed data curriculum remains effective even at a larger scale. The maintained robustness further suggests that Gaussian Noise Patching continues to offer benefits on more diverse datasets.
> Due to computational constraints, we were unable to conduct additional experiments on ImageNet-22K, but we plan to do so once we have access to sufficient resources.
>
> ### Question 3. How to combine masked image modelling and Gaussian noise patching is not clear. Does the GP applied to visible patches only? Or a replacement of masking?
> We thank the reviewer for bringing up this important aspect. To clarify, Gaussian noise is applied to a randomly selected region of the image with a random size to form a Gaussian noise patch. This process is independent of patch visibility in masked image modeling and is not intended as a replacement for masking. Instead, we introduce Gaussian Noise Patching as a complementary technique designed to help mitigate the frequency bias introduced by the data curriculum.
>
> ### Question 4. A similar strategy is popular in progressive training, which increases the resolution and the augmentation strength gradually during training [1,2]. As mentioned in [1], low-resolution does not mean an east task in MAE. In contrast, low-resolution images could act as regularisation to improve performance[3].
> We understand the reviewer’s question regarding the novelty of our work. One of our key contributions is a comprehensive analysis from the perspective of frequency and robustness, specifically focused on models trained with a low-frequency data curriculum. To the best of our knowledge, we are the first to identify and characterize the frequency bias introduced by such a training framework. Furthermore, we demonstrate that the combination of low-frequency training and Gaussian noise patching forms a non-trivial curriculum, which leads to more balanced and robust performance compared to using either technique in isolation. In addition to maintaining or improving downstream task performance and mitigating spurious correlations, our method achieves significant reductions in training time specifically for the DINOv2 model—an outcome not demonstrated in prior work.
> The reviewer refers to [1] and point out that low-resolution images act as a form of regularization rather than representing easier samples. From our perspective, this interpretation is not necessarily in conflict with defining them as “easy.” The self-attention mechanism in ViT inherently functions as a low-pass filter—it emphasizes low-frequency components while attenuating high-frequency signals. From this viewpoint, downsampled images align more closely with the model's internal bias and can thus be considered easier for the model to learn from.
>
> ### Question 5. Why use fewer training epochs? To show the speedup number, the final performance should match the original number.
> We thank the reviewer for pointing out this confusion. In our evaluation, the speedup is measured by comparing training time while keeping the final performance comparable. FastDINOv2 converges more quickly with the data curriculum, and in our experiments, it reaches the same level of accuracy as the baseline using fewer epochs. This faster convergence demonstrates the efficiency of our approach in reducing training cost while maintaining final performance.

---

> > ### Comment · Reviewer_keDD · 2025-08-05
> >
> > I recognise the original contribution on the study of low-frequency and high-frequency biases. However, this paper did not demonstrate enough results for the trade-off between efficiency and effectiveness. Typically, the speedup is calculated by measuring the training time gap for reaching a certain performance. Many papers present a figure of performance vs. training time or floaps, like [1]. It is crucial to show the trend of how the training strategy converges.
> >
> >
> > [1]: Leclerc G, Ilyas A, Engstrom L, et al. FFCV: Accelerating training by removing data bottlenecks[C]//Proceedings of the IEEE/CVF Conference on Computer Vision and Pattern Recognition. 2023: 12011-12020.

---

> ### Author Response · Authors · 2025-08-06
>
> We greatly appreciate your thoughtful follow-up. Upon reflection, we recognize the importance of figures illustrating convergence trends and comparing our method with the baseline. However, the current NeurIPS rebuttal policy prohibits posting links or images. Instead, we provide the following table comparing the training time required by DINOv2 and FastDINOv2 to reach specific linear probing accuracy levels. Training time is measured in NVIDIA L40S GPU hours:
>
> | Training Method | 55% Accuracy | 65% Accuracy | 75% Accuracy | 78% Accuracy |
> |-----------------|--------------|--------------|--------------|--------------|
> | DINOv2          | 2.6h         | 4.8h         | 12.48h       | 23.04h       |
> | FastDINOv2      | 2.64h        | 5.28h        | 11.5h        | 13.1h        |
>
> During the low-frequency training stage (when FastDINOv2 reaches 55% and 65% accuracy), FastDINOv2 achieves similar convergence speed to the baseline trained on full-resolution images. This is attributed to the inherent low-frequency bias in deep neural networks and transformers, such that the model is more inclined to learn low-frequency information first during training. Furthermore, this low-frequency stage provides a strong foundation for the subsequent full-resolution training stage (when FastDINOv2 reaches 75% and 78% accuracy), enabling FastDINOv2 to more efficiently and effectively learn fine-grained, high-resolution details from full-resolution images.
>
> We sincerely thank the reviewer for recommending this analysis—we believe this additional demonstration strengthens the contribution of our work. We will remain actively engaged in the remaining discussion period for additional comments.

---

> > ### Comment · Reviewer_keDD · 2025-08-07
> >
> > Thank you for your updated results. I will raise my rating but won't be high. The contribution to accelerate training is good, but the weakness are also clear. First, there are some theories to explain such behaviours during training that the model prefers low-frequency components at the beginning stage [1,2]. Since the model cannot learn low-frequency components, we can just remove high-frequency components to speed up the training, which has been applied in many studies. The paper lacks a comprehensive discussion about these theories and work. Second,  the real question is how to effectively extract high-frequency components. Although this paper proposed an easy yet effective solution by Gaussian Patching, it lacks enough experimental results to demonstrate its effectiveness. Third,  the Gaussian Patching does not sound an ultimate solution to enforce models to learning high-frequency components. In fact, there is relevant study on this [4].
> >
> >
> > [1] Balestriero R, LeCun Y. Learning by reconstruction produces uninformative features for perception[J]. arXiv preprint arXiv:2402.11337, 2024.
> > [2] Littwin E, Saremi O, Advani M, et al. How jepa avoids noisy features: The implicit bias of deep linear self distillation networks[J]. arXiv preprint arXiv:2407.03475, 2024.
> > [3] Wang H, Wu X, Huang Z, et al. High-frequency component helps explain the generalization of convolutional neural networks[C]//Proceedings of the IEEE/CVF conference on computer vision and pattern recognition. 2020: 8684-8694.
> > [4] Bizeul A, Sutter T M, Ryser A, et al. COMPONENTS BEAT PATCHES: EIGENVECTOR MASKING FOR VISUAL REPRESENTATION LEARNING[J].

---

> > > ### Author Response · Authors · 2025-08-07
> > >
> > > We appreciate your recognition of our contributions in this work. We acknowledge the lack of discussion on relevant literature and will incorporate additional analysis and citations, including the paper mentioned by the reviewer. We also recognize the importance of further demonstrating the effectiveness of Gaussian Patching through additional experiments and plan to explore more high-frequency extraction methods as part of our future work. We believe our submission offers meaningful value, as it presents a significant speedup for pretraining DINOv2—one of the most widely adopted self-supervised learning methods, with over 3 million downloads in the past month.
> > >
> > > Once again, we sincerely thank you for your constructive feedback and support!

---

### Official Review · Reviewer_Yv4e · 2025-07-02

**Clarity:** 4
**Significance:** 2
**Originality:** 3
**Rating:** 5
**Confidence:** 3

**Summary:**

This paper proposes FastDINOv2, a pre-training strategy for DINOv2. The core contribution is a two-stage, frequency-based curriculum that aims to accelerate training and improve robustness to common image corruptions. The curriculum begins by training the model exclusively on downsampled, low-frequency versions of images and then transitions to full resolution images augmented for the final 25% of training. The authors demonstrate on ImageNet that their method reduces training time, while maintaining competitive linear probing accuracy and comparable robustness on the ImageNet-C benchmark.

**Questions:**

I would not consider my current rating to be final, but this would be contingent on your answers to the below.

Could you provide any intuition or results from preliminary experiments that led to the choice of 75/25? How does the model perform with, for instance, a 50%/50%, 90%/10%, 0%/100% or 100%/0% splits? What led to your decision of 75/25?

In table 2 there is a fairly consistent trend, with one outlier. Do you have any insight into why model performance degrades as the image patches get smaller in all cases except the 128-224 which is worse than 112-224?

Do you think framing your model as competitive with the standard dinov2 performance is accurate? Or is your paper better framed as a slight trade off in accuracy but improvements in speed?

**Ethical Concerns:**

["NO or VERY MINOR ethics concerns only"]

**Final Justification:**

Updating my score in response to the rebuttal

**Limitations:**

Yes

**Quality:**

3

**Strengths And Weaknesses:**

Strengths:

The main strength of this work lies in its practical utility. The reported reduction in pre-training time from 16 days to 10 days are significant improvements. This directly addresses a major bottleneck in modern CV research and makes model training more attainable for smaller labs.

The paper's curriculum, which progresses from low-frequency easier examples to full-frequency harder examples, is intuitive and well-motivated. The authors identify that this curriculum creates a high-frequency bias and propose using Gaussian noise patching to create a more balanced model.

The authors evaluate their method across multiple ablations. They assess not only training efficiency but also performance on downstream tasks, ensuring representational quality beyond classification. Their robustness analysis on ImageNet-C breaks down performance by corruption type and frequency domain, which supports their claim.

Weaknesses:

The paper's main limitation, which the authors acknowledge, is the fixed schedule for the curriculum stages. The transition stages from low-resolution to full-resolution training after 75% of epochs is unmotivated. The paper lacks an ablation study on this ratio, making it difficult to assess how sensitive the final performance is to this critical hyperparameter or whether one stage is more important than the other. It is unclear if this 75/25 split is optimal.

The method is not a strict improvement. While achieving significant speed-ups, the final model for ImageNet-1K shows a slight drop in clean linear probing accuracy (77.8% for DINOv2 vs. 76.2% for FastDINOv2). The paper frames this as "competitive," but it is better framed as trade-off.

---

> ### Author Rebuttal · Authors · 2025-07-31
>
> We sincerely thank Reviewer Yv4e for the valuable comments and positive review. We would like to address each of your questions in detail:
>
> ---
>
> ### Question 1. Could you provide any intuition or results from preliminary experiments that led to the choice of 75/25? How does the model perform with, for instance, a 50%/50%, 90%/10%, 0%/100% or 100%/0% splits? What led to your decision of 75/25?
>
> We thank the reviewer for this essential question. For the ablation experiments on curriculum ratios, we used a fixed number of training epochs to ensure a fair comparison across different splits. Within this controlled training budget, our goal was to improve both learning efficiency and final performance through a two-stage curriculum.
> This design addresses two key aspects of our method: (a) leveraging the efficiency of low-resolution training, and (b) preserving final performance through progressive exposure to high-frequency content. The first stage uses downsampled (low-resolution) inputs that emphasize low-frequency information, while the second stage with high-resolution images enables learning of fine-grained details. If the first stage is too short, the model fails to fully benefit from this simplified setting. If it is too long, the model lacks sufficient exposure to the full-resolution features necessary for downstream performance.
> We experimented with multiple splits on ImageNet-100 with ViT-S, including 65/35, 75/25, and 85/15. Among these, the 75/25 split consistently yielded the best trade-off, achieving the highest linear probing accuracy. More extreme cases such as 0%/100% would have no acceleration effect, and 100%/0% would have only low-frequency data training, which performed notably worse than the above splits.
>
> | Training method | Split | Training Time (A6000 h) | Accuracy (ImageNet-100)  |
> |-----------------|-------|------------------------|-----------|
> | FastDINOv2      | 65/35 | 20.4                   | 77.6%     |
> | FastDINOv2      | 75/25 | 20.4                   | 78.1%     |
> | FastDINOv2      | 85/15 | 20.68                  | 76.8%     |
> | FastDINOv2      | 0/100 | 22.3                   | 73.9%     |
> | FastDINOv2      | 100/0 | 20.1                   | 69.7%     |
>
> This 75/25 curriculum design also enables cost-effective pretraining. By using downsampled images during the initial stage, GPU memory usage is significantly reduced, making it feasible to perform the majority of pretraining on smaller GPUs.
> | Training Method  | Training Epochs (Low-Res) | Training Epochs (Full-Res) | Max Memory Consumption (GB) Low-Res | Max Memory Consumption (GB) Full-Res |
> |------------------|---------------------------|----------------------------|------------------------------------|-------------------------------------|
> | DINOv2           | -                         | 200                        | -                                  | 33.5                                |
> | **FastDINOv2**   | 150                       | 50                         | 9.47                               | 33.5                                |
>
> ---
>
> ### Question 2. In table 2 there is a fairly consistent trend, with one outlier. Do you have any insight into why model performance degrades as the image patches get smaller in all cases except the 128-224 which is worse than 112-224?
>
> We thank the reviewer for this careful observation. Indeed, Table 2 shows that as the patch size of the low-resolution images decreases, model performance tends to degrade. This is expected, as smaller image patches discard more high-frequency information, limiting the model's ability to learn fine-grained features during the early phase of training.
> The 128-224 case performs slightly worse than 112-224, and we believe this minor deviation is due to a tradeoff between resolution and curriculum duration. Specifically, the 128-224 setting may not sufficiently reduce the input complexity to yield a significant efficiency benefit. In contrast, the 112-224 split achieves a better balance—providing just enough simplification to aid convergence while preserving meaningful structure.
>
> ---
>
> ### Question 3. Do you think framing your model as competitive with the standard dinov2 performance is accurate? Or is your paper better framed as a slight trade off in accuracy but improvements in speed?
>
> We thank the reviewer for this important clarification. Upon reflection, we agree that describing our model as “competitive” with the standard DINOv2 performance is somewhat overstated. One of our primary focuses lies in improving training speed and reducing computational resource requirements. Thus, it is more accurate to describe our model as a slight tradeoff rather than strictly competitive. We appreciate the reviewer pointing this out and will clarify this distinction more explicitly.

---

> > ### Comment · Reviewer_Yv4e · 2025-08-05
> > **Response**
> >
> > I appreciate the authors response and will be raising my score to a 5. While this paper is not groundbreaking I do think that it is worthwhile to be accepted.

---

> > > ### Author Response · Authors · 2025-08-06
> > >
> > > We appreciate your confirmation of our response. Thank you again for your valuable feedback and support!

---

### Official Review · Reviewer_PgPD · 2025-07-04

**Clarity:** 3
**Significance:** 3
**Originality:** 4
**Rating:** 5
**Confidence:** 3

**Summary:**

This paper introduces "FastDINOv2," a two-stage training curriculum for visual foundation models like DINOv2, which aims to improve training efficiency and model robustness simultaneously. The method begins with an initial training phase (75% of epochs) where the model is trained exclusively on down-sampled, low-resolution images to accelerate the learning of coarse, low-frequency features. In the second stage (the remaining 25% of epochs), the training transitions to full-resolution images and introduces a "Gaussian Noise Patching" augmentation. This augmentation is specifically designed to counteract the frequency bias induced in the first stage and to balance robustness across the entire frequency spectrum. Through extensive experiments, the authors demonstrate that their approach reduces training time by a factor of 1.6 and requires FLOPs by 2.25x, while maintaining competitive accuracy and robustness compared to the original DINOv2 model.

**Questions:**

- The fixed 75% / 25% split for the curriculum is a key design choice. Have you investigated other splits or considered adaptive strategies where the transition point is determined dynamically based on convergence metrics?
 - In the introduction (line 89) and elsewhere, you describe the curriculum as inducing a "high-frequency bias". This is not very clear. Would it be more precise to state that the model develops higher robustness to low-frequency corruptions, which implies a more substantial reliance on high-frequency features? Could you further elaborate on this causality?
 - Bicubic interpolation was used for downsampling the images. Did you study the impact of other low-pass filtering methods (e.g., a Gaussian blur before downsampling) on the efficiency and robustness of the final model?
 - Given the lack of error bars, would it be feasible to run at least a few training runs with different random seeds for a smaller configuration (e.g., ViT-S on ImageNet-100) to provide at least a sense of the variance and stability of the results?

**Ethical Concerns:**

["NO or VERY MINOR ethics concerns only"]

**Final Justification:**

I appreciate the author's response. I have increased my rating to 5 because the work contains a plenty of interesting aspects, so the paper is worthy to be accepted.

**Limitations:**

yes

**Paper Formatting Concerns:**

None.

**Quality:**

3

**Strengths And Weaknesses:**

**Strengths:**

1. The paper addresses a highly relevant problem of hugh computational costs in training foundation models. The achieved speedup are substantial.
2. The combination of a frequency-based curriculum with a targeted augmentation (Gaussian Patching) to compensate the induced bias is novel. The paper convincingly argues how robustness can be deliberately "built into" the training process rather than being merely a by-product of extreme scale.
3. The method is validated across various model and dataset sizes (ViT-S/B, ImageNet-100/1K). The evaluation is not limited to classification accuracy, but also includes robustness benchmarks (ImageNet-C), downstream tasks (semantic segmentation), and qualitative analyses (Grad-CAM, Fourier heatmaps), which makes the results very convincing.

**Weaknesses:**

1. The transition between the two training stages is set to a fixed value of 75% of the epochs. The authors themselves acknowledge that this heuristic approach may not be optimal. An adaptive mechanism would make the method more robust and generally applicable.
2. The results, particularly in the tables, are presented without error bars or other measures of statistical significance. The authors justify this in the checklist by citing high computational costs, which is understandable, but remains a scientific limitation as the variance of the results is unknown.
3. The term 'l40s' (e.g., in Abstract, Table 1) used to describe GPU resources is unclear and impairs readability. Clearly stating the GPU model (e.g., NVIDIA L40S) would enhance clarity.
4. The rationale for using downsampling as a proxy for low-frequency extraction is somewhat unclear or oversimplified. A more explicit discussion or comparison with standard low-pass filtering methods (e.g., Gaussian blur) would significantly strengthen this argument.

---

> ### Author Rebuttal · Authors · 2025-07-31
>
> We sincerely thank Reviewer PgPD for the positive review and important suggestions. Below, we would like to address each of your questions:
>
> ---
>
> ### Question 1. The fixed 75% / 25% split for the curriculum is a key design choice. Have you investigated other splits or considered adaptive strategies where the transition point is determined dynamically based on convergence metrics?
>
> We thank the reviewer for this important question and for recognizing the limitations involved in this design choice. We have indeed experimented with different splits, including 65/35, 75/25, and 85/15, and found that the 75/25 split yields the best performance.
> From a training convergence perspective, it is essential that the model can benefit sufficiently from both stages in our curriculum—low-frequency and full-resolution training. If the first stage is too short, the model may not develop a sufficiently strong initialization and gain the convergence speed up; if the second stage is too short, it may not fully learn more fine-grained details from training on high-resolution data.
> Given that the total number of training epochs is fixed in our current setting, we adopt a heuristic 75%/25% split for FastDINOv2. However, we agree that dynamically determining the transition point based on convergence metrics, such as online linear probing accuracy, could further optimize performance. We see strong potential in extending our method to include such adaptive curriculum strategies.
>
> | Training method | Split | Training Time (NVIDIA A6000 h) | Accuracy  |
> |-----------------|-------|------------------------|-----------|
> | FastDINOv2      | 65/35 | 20.4                   | 77.6%     |
> | FastDINOv2      | 75/25 | 20.4                   | 78.1%     |
> | FastDINOv2      | 85/15 | 20.68                  | 76.8%     |
>
> ---
>
> ### Question  2. In the introduction (line 89) and elsewhere, you describe the curriculum as inducing a "high-frequency bias". This is not very clear. Would it be more precise to state that the model develops higher robustness to low-frequency corruptions, which implies a more substantial reliance on high-frequency features? Could you further elaborate on this causality?
>
> We thank the reviewer for bringing up this thoughtful and important question. It is indeed more precise to describe the observed effect as increased robustness to low-frequency corruptions and a greater reliance on high-frequency features. This phenomenon arises from the intrinsic frequency biases of deep neural networks (DNNs), Vision Transformers (ViTs), and property of curriculum learning.
> Prior work has shown that DNNs exhibit a spectral bias—they tend to learn low-frequency components earlier in training and gradually incorporate higher-frequency features as training progresses [1,2]. Additionally, the self-attention mechanism in ViTs behaves like a low-pass filter [3], further reinforcing this early preference for low-frequency information.
> In our proposed curriculum, we exploit these inherent biases by explicitly organizing training to begin with low-frequency components, which serve as easier examples, and then transition to full-resolution images, which contain more high-frequency content and are thus harder. This progression aligns with findings in curriculum learning, where beginning with easier examples can smooth the optimization landscape and enable more effective learning of complex patterns later on [4].
> As a result, once the model has built a strong foundation from low-frequency features, it is better positioned to learn and rely on high-frequency components during the later stages of training. This leads to improved robustness against low-frequency corruptions, as the model has developed stronger sensitivity to the more informative high-frequency features.
>
> ---
>
> ### References
>
> [1] Rahaman, N., Baratin, A., Arpit, D., et al. (2019). On the Spectral Bias of Neural Networks. Proceedings of the 36th International Conference on Machine Learning (ICML).
> [2] Xu, Z.-Q. J., Zhang, Y., & Xiao, Y. (2019). Frequency principle: Fourier analysis sheds light on deep neural networks. arXiv preprint arXiv:1901.06523
> [3] Wang, Peihao, et al. Anti‑Oversmoothing in Deep Vision Transformers via the Fourier Domain Analysis: From Theory to Practice. ICLR 2022, arXiv:2203.05962, 2022.
> [4] Bengio, Y., Louradour, J., Collobert, R., & Weston, J. (2009). Curriculum learning. In Proceedings of the 26th Annual International Conference on Machine Learning (pp. 41–48). ACM.
>
> ---
>
> ### Question 3. Bicubic interpolation was used for downsampling the images. Did you study the impact of other low-pass filtering methods (e.g., a Gaussian blur before downsampling) on the efficiency and robustness of the final model?
>
> We thank the reviewer for this insightful question. Our initial hypothesis was that reducing high-frequency content in images could improve both convergence speed and model robustness, as low-frequency information typically captures the most generalizable features hence “easier”. To this end, we applied bicubic downsampling, which inherently acts as a low-pass filter and emphasizes low-frequency components.
> We also considered whether further reducing high-frequency signals with additional filtering such as Gaussian blur might improve performance. However, our experiments showed that bicubic downsampling alone was sufficient to retain the relevant low-frequency information while maintaining a balance with spatial detail. Additional filtering methods, including Gaussian blur and anti-aliasing, resulted in slight loss of detail and degraded model performance.
> Therefore, we concluded that downsampling with bicubic interpolation provides an effective trade-off between low-frequency extraction and information retention, and we adopted this approach in our final setup.
>
> | Filtering method           | Accuracy | Imagenet-C-100 (accuracy) |
> |---------------------------|----------|---------------------------|
> | Gaussian blur → Downsampling | 77.46%   | 51.24%                    |
> | Anti-Alias → Downsampling    | 76.5%    | 49.73%                    |
> | Downsampling                | 78.1%    | 52.48%                    |
>
> ---
>
> ### Question 4. Given the lack of error bars, would it be feasible to run at least a few training runs with different random seeds for a smaller configuration (e.g., ViT-S on ImageNet-100) to provide at least a sense of the variance and stability of the results?
>
> We thank the reviewer for understanding our limitation on computation resources and time. We conduct experiments using multiple random seeds (specifically, seeds 0, 10, and 5) on a smaller configuration—ViT-S trained on ImageNet-100 for FastDINOv2 with the 112-224 curriculum—and tested on ImageNet-100 validation set and ImageNet-100-C. While the output naturally varies across seeds, the overall performance remains consistent. These results support the robustness of our main findings.
>
> | Seed | Accuracy | Robustness |
> |-------|----------|------------|
> | 0     | 78.1%    | 52.48%     |
> | 10    | 77.43%   | 51.75%     |
> | 5     | 77.2%    | 51.68%     |

---

> > ### Author Response · Authors · 2025-08-09
> >
> > We hope our response has addressed all of your questions. Your feedbacks have been constructive and helpful for us in improving our paper. We sincerely thank you for the positive rating and look forward to addressing any additional comments or questions you may have!

---

### Official Review · Reviewer_PiGo · 2025-07-06

**Clarity:** 3
**Significance:** 2
**Originality:** 3
**Rating:** 4
**Confidence:** 3

**Summary:**

The paper introduces FastDINOv2, a two-stage curriculum training approach for DINOv2, which aims to enhance both training efficiency and robustness to corruption. The framework considers training first on downsampled images for 75% of the epochs, then changing to full-resolution, together with Gaussian noise patching, for the remaining 25% of the epochs. The authors demonstrate a 1.5x reduction in training time while maintaining competitive performance on (mostly) ImageNet-based benchmarks.

**Questions:**

I would appreciate it if the authors could address my concerns in the weaknesses section. I would consider increasing my score if those are addressed properly.

**Ethical Concerns:**

["NO or VERY MINOR ethics concerns only"]

**Final Justification:**

I appreciate the additional experiment provided in the rebuttal. After reading the other reviews and the rebuttal I decided to increase the score to borderline accept.
The proposed framework is simple and practical and shows promising results.
My main concern is still the lack of evaluation on larger datasets.

**Limitations:**

The authors mention some limitations in the conclusion of the paper.

**Paper Formatting Concerns:**

No format concerns

**Quality:**

2

**Strengths And Weaknesses:**

Strengths:
1. The approach achieves substantial acceleration without a major downgrade of results.
2. The frequency-based design of the curriculum is well-motivated and intuitive.
3. The authors provide insightful analysis, using Frequency-based analysis and Grad-CAM to support their claims.

Weaknesses:
1. Experiments are performed solely on DINOv2 (ViT). Will the curriculum show similar results for other methods? Is it also viable for CNNs?
2. Continuing the above, the experiments are performed mostly on an ImageNet subset (ImageNet-100). Raising concerns about how the curriculum performs on larger datasets (ImageNet-1K, ImageNet-22k, etc.)
3. The paper does not compare to competing frameworks.

---

> ### Author Rebuttal · Authors · 2025-07-31
>
> We sincerely thank the reviewer PiGo for their questions and comments. Below, we answer each question accordingly:
>
> ---
>
> ### 1. Experiments are performed solely on DINOv2 (ViT). Will the curriculum show similar results for other methods? Is it also viable for CNNs?
>
> We thank the reviewer for bringing up the important issue of generalization across frameworks and model architectures. To address this, we conducted additional experiments by training a SimCLR model with a ResNet backbone using our FastDINOv2 method (denoted as “FastSimCLR”). Using the LARS optimizer, the SimCLR baseline was trained for 200 epochs on full-resolution images from ImageNet-100, while FastSimCLR was trained for 150 epochs on low-frequency data followed by 50 epochs on full-resolution images.
> Despite the differences in inductive biases between ViTs and CNNs, FastSimCLR shows faster convergence. Furthermore, combining this curriculum with Gaussian noise patching improves robustness. We thank the reviewer again for this insightful suggestion–we believe this experiment further demonstrates the effectiveness of our method.
>
>
> | Training Method | Epochs | ImageNet-100 Acc | ImageNet-100-C Acc | Training Time (NVIDIA A6000 hours) |
> |-----------------|--------|----------|------------|-----------------------------|
> | SimCLR          | 200    | 64.48%   | 30.95%     | 14                          |
> | FastSimCLR      | 150+50 | 64.82%   | 38.73%     | 11.2                        |
>
> ---
>
>
> ### 2. The experiments are performed mostly on an ImageNet subset (ImageNet-100). Concerns about how the curriculum performs on larger datasets (ImageNet-1K, ImageNet-22K, etc.)
>
> We thank the reviewer for this insightful question. We agree that demonstrating the effectiveness of our method on larger datasets is crucial.
> Due to limitations in computational resources, we were unable to conduct experiments on ImageNet-22K. However, we did evaluate our approach on ImageNet-1K, as presented in Table 1. In this setting, we trained a ViT-B model on ImageNet-1K, evaluated its representations via linear probing on the ImageNet-1K validation set, and tested robustness on ImageNet-C.
> Our results show that FastDINOv2 achieves a substantial reduction in computational cost while maintaining performance comparable to the baseline DINOv2. Notably, the improvements in convergence speed and robustness observed on ImageNet-100 carry over to ImageNet-1K, suggesting that our curriculum design generalizes well as dataset scale increases. These findings imply that our method may also scale effectively to even larger datasets such as ImageNet-22K. With additional computational resources, we intend to validate our approach at the ImageNet-22K scale.
>
> ---
>
> ### 3. The paper does not compare to competing frameworks.
>
> We appreciate the reviewer’s concern regarding comparisons to other efficient training frameworks. Many existing approaches are tailored to specific models—such as [1] or [2]—and cannot be applied directly to DINOv2 without significant changes to training configurations. Moreover, more general frameworks like [3] do not observe a speed-up for DINO in their experiments. To our knowledge, our work is the first to propose an efficient training pipeline for DINOv2 based on a data curriculum strategy.
>
> ---
>
> ### References
>
> 1. Li, Runze, et al. *RECLIP: Resource-Efficient CLIP by Training with Small Images*. arXiv preprint arXiv:2304.06028, 2023.
>
> 2. Wu, Jiantao, et al. *DailyMAE: Towards Pretraining Masked Autoencoders in One Day*. arXiv preprint arXiv:2404.00509, 2024.
>
> 3. Wang, Yulin, et al. *EfficientTrain++: Generalized Curriculum Learning for Efficient Visual Backbone Training*. *IEEE Transactions on Pattern Analysis and Machine Intelligence*, vol. 46, no. 12, 2024, pp. 8036–8055.

---

> > ### Comment · Reviewer_PiGo · 2025-08-06
> >
> > I appreciate the additional experiment provided in the rebuttal. After reading the other reviews and the rebuttal I decided to increase the score to borderline accept.
> > The proposed framework is simple and practical and shows promising results.
> > My main concern is still the lack of evaluation on larger datasets.

---

> > > ### Author Response · Authors · 2025-08-06
> > >
> > > We appreciate your thoughtful consideration of our responses and are grateful for your support! We acknowledge the concern regarding evaluation on larger datasets and regard it as important feedback for further validating and strengthening our work.

---

### Note · Authors · 2025-08-12

Dear AC and reviewers,

As a final remark, we sincerely thank you for your time, engagement, and constructive feedback, which have been instrumental in improving our work. We hope that the additional experiments provide stronger evidence that our design choices for low-frequency components filtering and training stage splits outperform other available options, while maintaining generalization ability across downstream tasks, architectures and training methods. We believe these results further strengthen our submission and highlight its potential to benefit a broad community by accelerating pretraining for DINOv2—one of the most widely used self-supervised learning methods. We will incorporate these results, along with expanded discussions of relevant literature, to further enhance the work.
Thank you all again!

---

### Decision · Program_Chairs · 2025-09-17

**Decision:**

Accept (poster)

**Comment:**

The paper proposes FastDINOv2, a two-stage frequency-based curriculum to accelerate DINOv2 pre-training. The method trains first on low-resolution images, then transitions to full-resolution images with a novel "Gaussian Noise Patching" augmentation, claiming a 1.6x training speedup while maintaining competitive accuracy and improving robustness.

The primary strength of this work is its practical impact, offering a significant reduction in the computational cost of training foundation models. The curriculum is well-motivated and its combination with a targeted noise augmentation to improve robustness is a novel contribution, supported by a thorough evaluation covering accuracy, robustness, and downstream tasks. The main weaknesses are that the method presents a trade-off—a minor drop in clean accuracy in exchange for speed—and its evaluation is limited to ImageNet-scale datasets. The paper could also better situate itself within the existing literature on spectral bias.

Despite these points, I recommend acceptance due to the paper's clear practical value in making foundation model training more efficient and robust. The discussion period was crucial in reaching this decision. Initial reviewer concerns about the method's generalizability, hyperparameter choices, and performance framing were addressed directly by the authors with new experiments and clarifications. These responses caused three of four reviewers to raise their scores, solidifying a strong consensus for acceptance. The strengths and practical utility of the work clearly outweigh its remaining limitations.